# Parameter Estimation, Robust Controller Design and Performance Analysis for an Electric Power Steering System

**Van Giao Nguyen [1]** , **Xuexun Guo [1,\*]**, **Chengcai Zhang [1]** and **Xuan Khoa Tran [2]**

[1]  School of Automotive Engineering, Wuhan University of Technology, Wuhan 430070, China;
   quocgiao82@gmail.com (V.G.N.); zhangchc@163.com (C.Z.)
[2]  School of Automation, Wuhan University of Technology, Wuhan 430070, China; khoadkt@gmail.com
\*  Correspondence: guoxx@whut.edu.cn; Tel.: +86-27-87858005

**Abstract:** This paper presents a parameter estimation, robust controller design and performance analysis for an electric power steering (EPS) system. The parametrical analysis includes the EPS parameters and disturbances, such as the assist motor parameters, sensor-measurement noise, and random road factors, allowing the EPS stability to be extensively investigated. Based on the loop-shaping technique, the system controller is designed to increase the EPS stability and performance. The loop-shaping procedure is proposed to minimize the influence of system disturbances on the system outputs. The simplified refined instrumental variable (SRIV) algorithm, least squares state variable filter (LSSVF) algorithm and instrumental variable state variable filter (IVSVF) algorithm are applied to reduce the model mismatching between the theoretical EPS models and the real EPS model, as the EPS parameters can be accurately identified based on the experimental EPS data. The performance of the proposed method is thus compared to that of the proportional-integral-derivative (PID) test bench results for the EPS system. The experimental results demonstrated that the proposed loop-shaping controller provides good tracking performance while ensuring the stability of the EPS system.

**Keywords:** electric power steering; loop-shaping control; model identification; simplified refined instrumental variable; least squares state variable filter; instrumental variable state variable filter

## 1. Introduction

In grounded vehicles, the steering system plays an important role in ensuring the motion trajectory of the car according to the driver's desires and road traffic. Hydraulic power steering systems powered by the engine were introduced to reduce driver torque on the steering wheel. In recent years, EPS (electric power steering) systems have been widely deployed in modern vehicles due to their noticeable advantages over hydraulic power steering systems, including higher efficiency, more compact structure, better maintainability and lower fuel consumption [1–5]. In EPS systems, the assist torque is generated by an electric motor, which is powered by the car battery. Thus, the control strategies of assist torque need to consider the torque adjustment in accordance with the driver input torque, driver steering feeling and road conditions. Several control algorithms of EPS systems have been investigated for EPS control, including PI and PID EPS controllers [6–9]. Although PI-and PID-controlled EPS exhibited satisfactory performance, their performance can be further improved by considering model mismatching and disturbances, including friction and tire forces. To this end, EPS-based fuzzy controllers have been proposed to reduce the influence of friction and road disturbances [10,11]. Wang [12] designed a robust $H_\infty$ output-feedback yaw control algorithm based on differential steering and the complete failure of the active front-wheel steering. Moradkhani [13] introduced a loop-shaping

technique to control the steering-column torque, which is measured by the torque sensor and reference torque tracking neglecting the estimation of the EPS parameters. Gao [14] proposed a fault-tolerant control strategy based on fault estimation and reconstruction. Ma [15] proposed an active disturbance rejection approach for the EPS system, which can significantly reduce steering wheel vibration torque. Lee [16] introduced a robust steering-assist torque control of EPS systems that achieved optimal steering wheel torque tracking performance. Yang [17] presented a new control framework for vehicle EPS systems based on admittance control, which can augment the base steering feel and improve the road condition awareness. Zhang [18] proposed a nonlinear decoupling control method to improve the stability and maneuverability of EPS.

The robustness of EPS control methods is especially important under high assist gain. In previous studies on this topic, the linear quadratic regulator (LQR) approach was adopted to attain high robustness and stability [19,20], in turn achieving high phase and gain margins. Further optimal controllers based on the LQR-based EPS system were proposed in [21–23]. System identification of the control plant is essential in model-based control strategies. Meanwhile, the control performance will be notably improved if the control plant parameters are identified in a precise manner. Thus, system identification is important in the design of EPS control systems. Several methods have been proposed to identify and determine mathematical models for control plants, including the off-line identification method for identifying the brushless DC parameters [24]. Another system-identification strategy consists of the crack and bearing dynamic parameters, which was investigated in [25]. Recently, Zhang [26] designed a controller by implementing an observer based on the linear-parameter-varying (LPV) model. The seating-system model and a 5 degrees of freedom (5-DOF) model were developed in [27] for optimal vibration control. Furthermore, Zhang [28] investigated the uncertainty of scheduling parameters for practical applications.

Most existing EPS controllers do not consider EPS parameter estimation with a pre-determined EPS mathematical model. As a result, their control performance is compromised in practical EPS implementation. The reason for this is that the practical model parameters vary due to road disturbances, measurement noise and other factors. To reduce the model mismatch between the theoretical EPS models and EPS plant, this paper identifies each EPS parameter using three different algorithms and then designs a controller based on the system-identification experiments. In addition, this paper investigates sampling rate, which significantly affects the identification results, and finds the optimal sampling rate based on experiments. Based on evaluated the estimation criterion in this paper, the LSSVF algorithm is the simplest method for system parameter estimation. The IVSVF algorithm reduces the bias of the LSSVF algorithm, whereas the SRIV algorithm is the optimal estimation method. Hence, the SRIV algorithm is adopted for estimating the parameters of the control plants. In this algorithm, an adaptive procedure is employed. The SRIV algorithm is shown to be highly robust in practical applications. Hence, the model can provide optimal system parameters considering the measurement noise, thus making the nominal model of the EPS system highly accurate. However, the estimated model of EPS system is the nominal mode. The EPS system controller is challenging design problem. The EPS system controller must provide good tracking performance to ensure system stability and good steering feel. The proposed loop-shaping controller in this paper satisfies these several objectives.

This paper is organized as follows. Section 2 introduces the operational principle and mathematical model of EPS system. Section 3 introduces the mathematical basis used to implement the estimation algorithms. Section 4 presents an identification analysis of the motor parameters. Section 5 includes both the simulation identification and experiment identification of the proposed EPS system. Section 6 presents a design of the loop-shaping controller. Section 7 shows the EPS test bench. The verification of the EPS control algorithms is presented in Section 8.

## 2. Operational Principle and Mathematical Model

This section introduces the C-type EPS model and its operational principles. The dynamic equations of the EPS system are also presented based on Newton's second law.

### 2.1. Operational Principle

Figure 1 illustrates the structure of a typical EPS system; the EPS consists of a steering wheel, torque sensor, vehicle velocity sensor, electronic control unit (ECU), electrical motor and rack and pinion assembly [3]. The driver torque and vehicle velocity are sent to the ECU to calculate the appropriate value of the assist torque using the desired assist torque curve. The assist torque is provided by the electric motor and transmission mechanism.

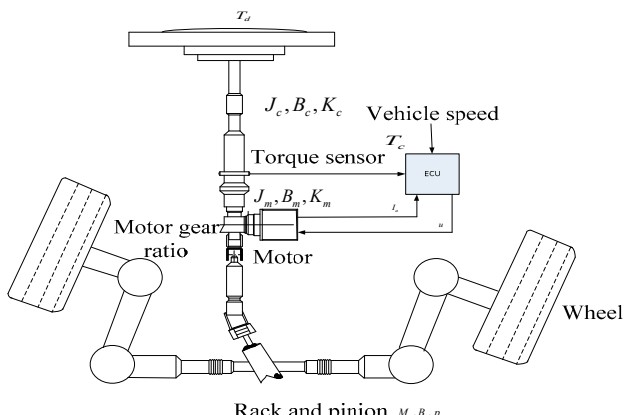

**Figure 1.** Electric power steering system model.

The assist torque is needed to overcome the steering resistance and complete the turning action while reducing the driver applied torque. The main objective of the controller is to ensure that the curve of the desired driver torque always tracks the actual driver torque at different vehicle speeds.

### 2.2. Mathematical Model

According to Newton's second law, the dynamical model of typical EPS systems is described below.

The column dynamics are described as follows:

$$J_c\ddot{\theta}_c = T_d - B_c\dot{\theta}_c - K_c(\theta_c - \frac{p_r}{r_p}) \tag{1}$$

where $\theta_c$, $r_p$, $B_c$, $K_c$, $J_c$, $p_r$, $T_d$ are the steering wheel angular position, pinion radius, steering wheel damping, steering column torsional stiffness, steering wheel moment of inertia, steering rack displacement and driver torque, respectively.

The motor mechanical dynamics are described by the following equation:

$$J_m\ddot{\theta}_m = k_e I_a - B_m\dot{\theta}_m - K_m(\theta_m - \frac{p_r}{r_p}N) \tag{2}$$

where $J_m$, $\theta_m$, $k_e$, $B_m$, $N$ are the motor moment of inertia motor torque of inertia, motor column angular position, motor torque and voltage constant, motor and gearbox damping, motor and gearbox torsional stiffness and motor gear ratio, respectively.

The rack dynamics are described by the following equation:

$$M_r\ddot{p}_r = \frac{K_c}{r_p}(\theta_c - \frac{p_r}{r_p}) - B_r\dot{p}_r + \frac{K_m N}{r_p}(\theta_m - \frac{p_r}{r_p}N) - K_r p_r \tag{3}$$

where $M_r$, $K_r$, $B_r$ are the rack and wheel assembly mass, the tire or rack centering spring rate and the rack damping, respectively.

Equations (1)–(3) are also provided for the representation of the EPS model [21]. In addition, based on Kirchhoff's law, the assist motor of the EPS system can be described as follows:

$$u = R_m I_a + L_m \dot{I}_a + k_e \dot{\theta}_m \tag{4}$$

where $R_m$, $L_m$ are the motor resistance and motor inductance, respectively, and $u$, $k_e$ and $I_a$ are the voltage, voltage constant and current of the motor armature, respectively.

Based on Equation (4), the generalized transfer function of the motor can be described as follows:

$$G_0(s) = \frac{I(s)}{U(s)} = \frac{1}{L} \frac{s + \frac{B_m}{J_m}}{s^2 + s\left(\frac{R_m}{L_m} + \frac{B_m}{J_m}\right) + \frac{R_m}{L_m}\left(\frac{M_m}{J_m} + k_e^2\right)} = \frac{cs + d}{(s + a)(s + b)} \tag{5}$$

where

$$a = \frac{1}{2}\left(\frac{R_m J_m + L_m B_m}{L_m J_m} - \sqrt{\left(\frac{R_m J_m + L_m B_m}{L_m J_m}\right)^2 - 4\frac{R_m B_m + k_e^2}{L_m J_m}}\right),$$
$$b = \frac{1}{2}\left(\frac{R_m J_m + L_m B_m}{L_m J_m} + \sqrt{\left(\frac{R_m J_m + L_m B_m}{L_m J_m}\right)^2 - 4\frac{R_m B_m + k_e^2}{L_m J_m}}\right),$$
$$c = \frac{1}{L_m}, \ d = \frac{B_m}{L_m J_m}$$

The equation of the torque sensor is described as follows:

$$T_c = K_c\left(\theta_c - \frac{p_r}{r_p}\right) \tag{6}$$

The torque equation of the assist motor is described as follows:

$$T_a = N k_e I_a \tag{7}$$

To simply the identification of the EPS parameters, the assist motor is initially not considered. Considering Equations (1)–(3), with regard to the Laplace transformation, the following equations can be established:

$$T_d(s) = J_c s^2 \theta_c(s) + B_c s \theta_c(s) + K_c \theta_c(s) - K_c \frac{p_r(s)}{r_p} \tag{8}$$

$$M_r s^2 p_r(s) + B_r s p_r(s) = \frac{K_c}{r_p}\theta_c(s) - \frac{K_c}{r_p^2}p_r(s) - K_r p_r(s) \tag{9}$$

where $T_d(s)$, $\theta_c(s)$, $p_r(s)$ are driver torque, the steering wheel angular position, and steering rack displacement in the Laplace $s$ domain, respectively.

Thus, the transfer function of the EPS system can be obtained as follows:

$$G_1(s) = \frac{p_r(s)}{T_d(s)} = \frac{1}{\frac{r_p}{K_c}(J_c s^2 + B_c s K_c)(M_r s^2 + B_r s + \frac{K_c}{r_p^2} + K_r) - \frac{K_c}{r_p}} = \frac{a_0}{s^4 + b_3 s^3 + b_2 s^2 + b_1 s + b_0} \tag{10}$$

where

$$a_0 = \frac{K_c}{r_p J_c M_r}, \ b_0 = \frac{1}{J_c M_r}\left(\frac{K_c^2}{r_p^2} + K_c K_r\right) - \frac{K_c^2}{J_c M_r r_p^2},$$

$$b_1 = \frac{1}{J_c M_r}\left[B_c\left(\frac{K_c}{r_p^2} + K_r\right) + K_c K_r\right], \ b_2 = \frac{1}{J_c M_r}\left[J_c\left(\frac{K_c}{r_p^2} + K_r\right) + B_c B_r + K_c M_r\right], \ b_3 = \frac{J_c B_r + M_r B_c}{J_c M_r}$$

## 3. Background of the Algorithms

As mentioned above, it is essential to estimate the parameters of a plant based on both the input and output data. This paper proposes three algorithms to estimate the parameters of the motor and EPS system, and then compares their performance for choosing the optimal algorithm.

Based on a fractional differential equation, the fractional mathematical model is written as follows:

$$y(t) + a_1 H^{\alpha_1} y(t) + \ldots + a_{m_A} H^{\alpha_{m_A}} y(t) = b_0 H^{\beta_0} u(t) + b_1 H^{\beta_1} u(t) + \ldots + b_{m_B} H^{\beta_{m_B}} u(t) \quad (11)$$

where $\alpha_1 < \alpha_2 < \ldots < \alpha_{m_A}$, $\beta_0 < \beta_1 < \ldots < \beta_{m_B}$ are the differentiation orders, $u(t)$ and $y(t)$ are the input signal and output signal, respectively, and $H^\gamma = \left(\frac{d}{dt}\right)^\gamma$ is the differentiation to an arbitrary order. The measured noise $p(t)$ is added to the measured output signal as below:

$$y^*(t) = y(t) + p(t) \quad (12)$$

The error can be written as follows:

$$e(t) = y^*(t) - \psi^*(t)^T \varphi \quad (13)$$

where

$$\psi^*(t)^T = \left[ -H^{\alpha_1} y^*(t), \ldots, -H^{\alpha_{m_A}} y^*(t), H^{\beta_0} u(t), \ldots, H^{\beta_{m_B}} u(t) \right] \quad (14)$$

Hence, the estimated parameters of the model are defined as follows:

$$\varphi^T = [a_1, \ldots, a_{m_A}, b_0, b_1, \ldots, b_{m_B}] \quad (15)$$

Achieving model parameters that are close to the real plant parameters requires that the sum of the error squares is as small as possible. Hence, the least squares algorithm can be written as follows:

$$\hat{\varphi}_{LS} = \left[ \int_0^T \psi^*(t)^T \psi^*(t) dt \right]^{-1} \int_0^T \psi^*(t)^T y^*(t) dt \quad (16)$$

The input signal and output signal observed at regular sample $T_s, 2T_s, \ldots, KT_s$. Hence, the parameters estimation can be an approximated as follows:

$$\hat{\varphi}_{LS} = \left( \phi^{*T} \phi^* \right)^{-1} \phi^{*T} Y \quad (17)$$

where

$$\phi^* = \left( \psi^*(T_s), \ \psi^*(2T_s), \ldots, \psi^*(KT_s) \right)^T \quad (18)$$

The real system always includes noise, so estimating the parameters considering noise is very important. The noise output is used by direct fractional differentiations lead inaccurate results. Hence, Poisson's filters can be adopted to be applied to the input and the output signals [29]. The output signal of the filter can be written as follows:

$$\begin{cases} H^{\beta_i} u_f = u(t) \otimes L^{-1} \{ F_{\beta_i}(s) \} \\ H^{\alpha_j} y_f^* = y(t) \otimes L^{-1} \{ F_{\alpha_j}(s) \} \end{cases} \quad (19)$$

where $L^{-1}$ denotes for the inverse Laplace transform. The $\otimes$ symbol is the convolution operator. The variable $(\cdot)_f$ refers to the used filter.

The error $e_f(t)$ can be written as follows:

$$e_f(t) = y_f^*(t) - \psi_f^*(t)^T \varphi \quad (20)$$

where

$$\psi_f^*(t)^T = \left[ -H^{\alpha_1} y_f^*(t), \ldots, -H^{\alpha_{m_A}} y_f^*(t), H^{\beta_0} u_f(t), \ldots, H^{\beta_{m_B}} u_f(t) \right] \quad (21)$$

where the variable $(\cdot)^T$ denotes the transposition.

The minimum sum of squared errors $e_f(t)$ leads to the estimation of parameters using least squares state variable filter algorithm, which can be approximated as follows:

$$\underset{LSSVF}{\overset{\wedge}{\varphi}} = (\phi_f^{*T}\phi_f^*)^{-1}\phi_f^{*T}Y_f^* \tag{22}$$

where

$$\phi_f^* = (\psi_f^*(T_s), \; \psi_f^*(2T_s), \ldots, \psi_f^*(KT_s))^T \tag{23}$$

Based on an instrumental regression, the following can be written:

$$\psi_f^{IV}(t)^T = \left[-H^{\alpha_1}y_f^{IV}(t),\ldots,-H^{\alpha_{m_A}}y_f^{IV}(t),-H^{\beta_0}u_f^{IV}(t),\ldots,-H^{\beta_{m_B}}u_f^{IV}(t)\right] \tag{24}$$

where the variable $(\cdot)^{IV}$ defines the used instrumental variable.

The estimated parameters using $IV$ state variable filter algorithm are approximated as follows:

$$\underset{IVSVF}{\overset{\wedge}{\varphi}} = (\phi_f^{IV^T}\phi_f^*)^{-1}\phi_f^{IV^T}Y_f^* \tag{25}$$

where

$$\phi_f^{IV} = (\phi_f^{IV}(T_s), \; \phi_f^{IV}(2T_s), \ldots, \phi_f^{IV}(KT_s))^T \tag{26}$$

When Poisson's filters are changed by the optimal filter introduced in [30], the optimal regression vector can be written:

$$\psi_f^{opt}(t)^T = \left[-H^{\alpha_1}y_f(t),\ldots,-H^{\alpha_m}y_f(t),H^{\beta_0}u_f(t),\ldots,H^{\beta_{m_B}}u_f(t)\right] \tag{27}$$

where the variable $(\cdot)^{opt}$ defines the optimum.

The filter will be updated with the new estimated parameters. Then, the pre-filter derivatives of $u(t)$, $y(t)$ and $y^{IV}(t)$ are written as in [29]. Hence, the regression vectors $\psi_f^*(t)^T$ and $\psi_f^{IV}(t)^T$ are generated as in Equations (21) and (24). Finally, the estimated parameters are calculated at each iteration as follows:

$$\varphi_{SRIV}^{iter} = (\phi_f^{IV^T}\phi_f^*)^{-1}\phi_f^{IV^T}Y_f^* \tag{28}$$

where the variable $(\cdot)^{iter}$ defines the iteration.

The iteration will stop when the estimated parameters reach optimal values.

## 4. Identification Analysis of the Motor Parameters

### 4.1. Simulation Identification

The motor parameter values used to simulate the identification algorithms are shown in Table 1.

**Table 1.** Motor parameters used in the simulations.

| Motor Parameters | Value | Unit |
|---|---|---|
| Armature resistance $R_m$ | 0.35 | $\Omega$ |
| Armature inductance $L_m$ | 0.001 | H |
| Motor back EMF constant $K_t$ | 0.0433 | V·s |
| Motor rotor inertia $J_m$ | 0.00018 | kg·m$^2$ |
| Hydraulic friction coefficient of Motor rotor $B_m$ | 0.0034 | N·m·s |

The following transfer functions can be calculated by combining the motor transfer function in Equation (5) and Table 1:

$$G_0(s) = \frac{1000s + 18889}{s^2 + 368.8889s + 6611.1} \tag{29}$$

The sampling rate is very important for parameter estimation. A sampling rate of 800 Hz is the optimal sampling rate in this paper, as it satisfies the Nyquist theorem, allowing high precision in model parameter identification. In addition, the experiments were implemented at the different sampling rates in order to adopt the optimal sampling rate. From the above basic theoretical analysis of the proposed the algorithms, Equation (28) implemented in the MATLAB is used to estimate the motor transfer function at different sampling rates. The simulation results for the SRIV algorithm with a signal-to-noise ratio (SNR) of 30 dB and using different sampling rates are shown in Figure 2. The Bode plots using individual sampling rates of 300 Hz, 800 Hz and 10 kHz compared with the actual model are shown in Figure 2.

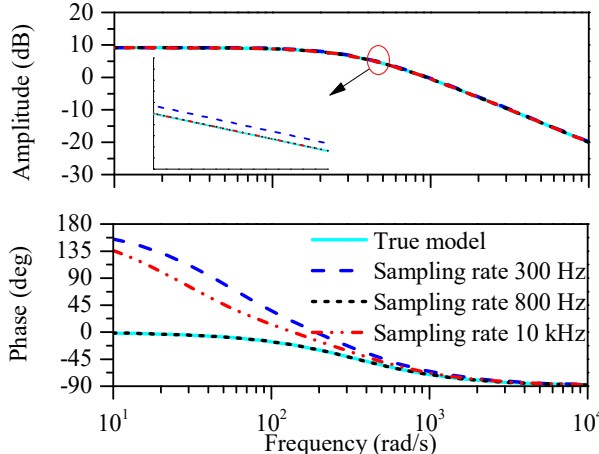

**Figure 2.** Bode plots of the SRIV identification algorithm with a signal-to-noise ratio of 30 dB.

A sampling rate of 300 Hz is too low to satisfy the Nyquist sampling theorem, and thus, the identification effect is not ideal. A sampling rate of 10 kHz satisfies the Nyquist sampling theorem but is too high, leading to a waste of memory space and long signal processing times. Using high-performance filters to filter the noise in the high frequency range leads to a high degree of control over system implementation cost. In addition, high-quality filters, fast processor speed and large memory capacity are necessary to respond to the high sampling rate. Without these, undesirable results, such as incorrect model parameter identification, would be obtained. According to this analysis, phase frequency characteristics of 300 Hz or 10 kHz are quite different from the actual one. In the simulation identification, the noises added to the system outputs are uncertain, and thus, the simulation outputs are different for every trial. The results also show that the identification result is unstable when the sampling rate is high. Using Equations (17), (22) and (28), the motor transfer function was estimated by three different algorithms. The Bode plots of the identification results using the SRIV, IVSVF and LSSVF algorithms are shown in Figure 3. These algorithms are appropriate for the actual model. The LSSVF algorithm is the simplest method for system parameter identification. The IVSVF algorithm reduces the bias of the LSSVF algorithm, whereas the SRIV algorithm is the optimal identification algorithm. Hence, the Bode plot of the SRIV algorithm is the most appropriate for the actual model, and the results obtained using the IVSVF algorithm are better than those obtained using the LSSVF algorithm, as confirmed in [31,32].

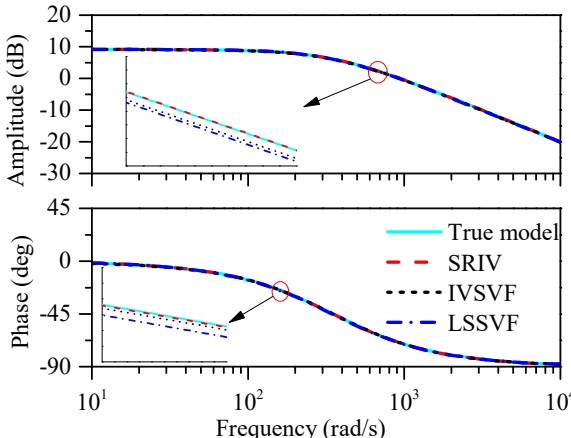

**Figure 3.** Bode plots of the proposed identification algorithms compared with that of the actual model.

To select an appropriate model structure, we utilized various criteria, such as the coefficient of determination (RT2), Young's information criterion (YIC), the condition number (Cond), the Akaike information criterion (AIC), and *nb*, *nf*, and *nk* denote the number of numerator and denominator parameters and number of samples for the delay of the system, which are shown in Table 2, in accordance with [31,32].

**Table 2.** SRIV identification algorithm factors.

| *nb* | *nf* | *nk* | RT2 | YIC | Cond | AIC |
|---|---|---|---|---|---|---|
| 1 | 1 | 0 | 0.99891 | −10.831 | 27.293 | −5.1153 |
| 1 | 4 | 0 | 0.96856 | −10.557 | $3.7446 \times 10^9$ | −1.7471 |
| 3 | 3 | 0 | 0.99893 | −9.4845 | $5.0471 \times 10^7$ | −5.1219 |
| 2 | 1 | 0 | 0.99891 | −6.8284 | 1170.7 | −5.1136 |
| 2 | 2 | 0 | 0.99897 | −11.3369 | $5.1443 \times 10^5$ | −5.1377 |

Figure 4 shows the inputs of the motor during the identification simulation. The input signal is a pseudo random binary sequence (PRBS).

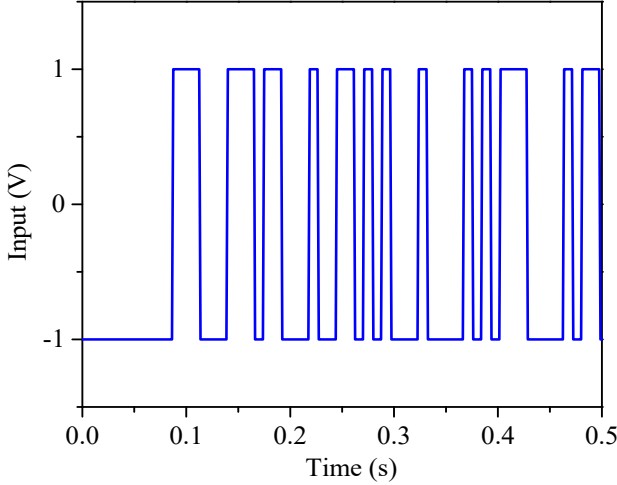

**Figure 4.** Inputs of the motor during the identification simulation.

Figure 5 shows a comparison of the simulation data and test data using the SRIV algorithm when *nb* and *nf* are both set to 2. The test data change roughly around the maximum and minimum positions due to the test noise. The simulation data go through these positions smoothly. Ignoring the

effect of the noise, the test data and simulation coincide, demonstrating the effectiveness of the SRIV identification algorithm.

The analysis of the motor simulations illustrates that when the identification conditions include a sampling rate of 800 Hz and *nb* and *nf* both set to 2, the desired motor model can be obtained through the three identification algorithms. However, the most effective algorithm is the SRIV algorithm, followed by the IVSVF algorithm. The final identification algorithm cannot be determined-based solely on the analysis of the simulation results; experiments must also be considered.

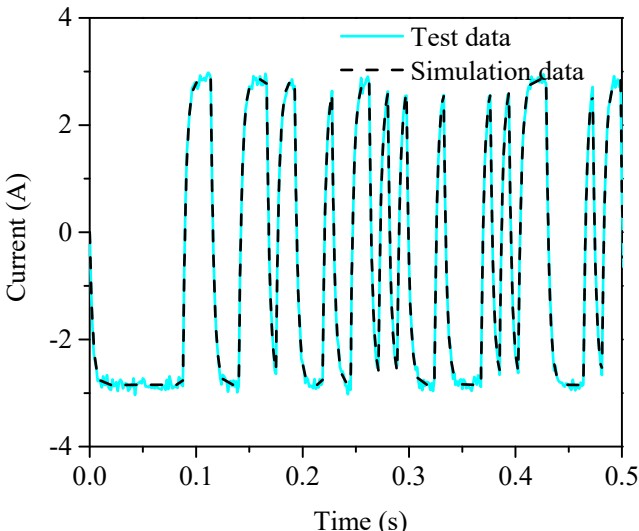

**Figure 5.** Comparison of the simulation data and test data using the SRIV algorithm.

### 4.2. Identification Experiment

In this subsection, the identification experiment for the motor parameters based on the recorded voltage and current of the motor is presented. The mathematical model of the motor is obtained using different identification algorithms with different conditions. Before the motor identification experiment, the load motor is shortened, and the electromagnetic clutch of the load motor is supplied with a 24 V direct source, so it can provide the load torque when the assist motor is rotated. In the experiment, when the starting command is received by the upper computer, the PRBS in the motor control module will circulate until the end of the experiment. At the same time, the data acquisition system will measure the current, the terminal voltage of the assist motor and the load current. The upper computer can change the amplitude and the clock time of the PRBS in the motor control module through the CAN bus, and can also change the sampling rate of the data acquisition system.

Figure 6 highlights the step response for identifying the motor model using different sampling rates. The conditions for the sampling rates of 1 kHz and 900 Hz are 20% and 50% pulse width modulation (PWM) duty cycle (motor only rotates clockwise) with a time of 20 ms, respectively. When the sampling rate is 800 Hz, the identification condition of the model is 60% (positive and negative, clockwise and anticlockwise) PWM duty cycle. The identification model yields the most accurate identification under this condition. The step response of the motor model, identified at different sampling rates, takes 20 ms to reach steady-state mode. The step response curve of the motor model identified by the experiments is also analyzed.

Considering the transfer function of the motor, there are five motor parameters: the motor resistance, inductance, back EMF constant, rotor inertia and rotor friction coefficient. Only 4 parameters can be identified in the motor model structure when *nb* and *nf* are equal to 2. Hence, in the identification test, only two parameters of the motor (the motor resistance and inductance) can be identified; the other parameters can be obtained through an identification experiment in which the motor voltage is the input and the motor speed is the output. Table 3 shows that when the sampling rate is too

high, the parameters of the motor are clearly not consistent with the actual physical parameters of the motor. Therefore, a reasonable choice of sampling rate is critical for an accurate motor parameter identification. The results show the motor resistance (R = 0.281) and inductance (L = 0.0015).

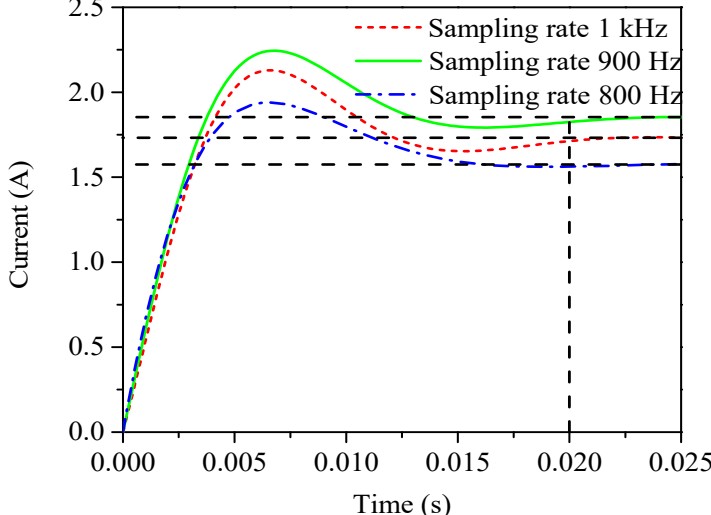

**Figure 6.** Step response for identifying the motor model at different sampling rates.

**Table 3.** Identification of the motor parameters under different identification conditions.

| Identification Results | Motor Parameter | | | |
|---|---|---|---|---|
| Sampling Rate (Hz) | Amplitude (%) | Clock Time (ms) | Resistance (Ω) | Inductance (H) |
| 10k | 20 50 | 200 | −0.374 | 0.0020 |
| 5k | 20 50 | 200 | −0.759 | 0.0023 |
| 1k | 20 60 | 200 | 0.096 | 0.0015 |
| 800 | 20 60 | 200 | 0.101 | 0.0014 |
| 800 | −60 60 | 500 | 0.281 | 0.0015 |

Figure 7 compares the experimental and identification results at a sampling rate of 800 Hz. When the sampling rate is 800 Hz, the motor model parameters can be obtained with a high level of accuracy.

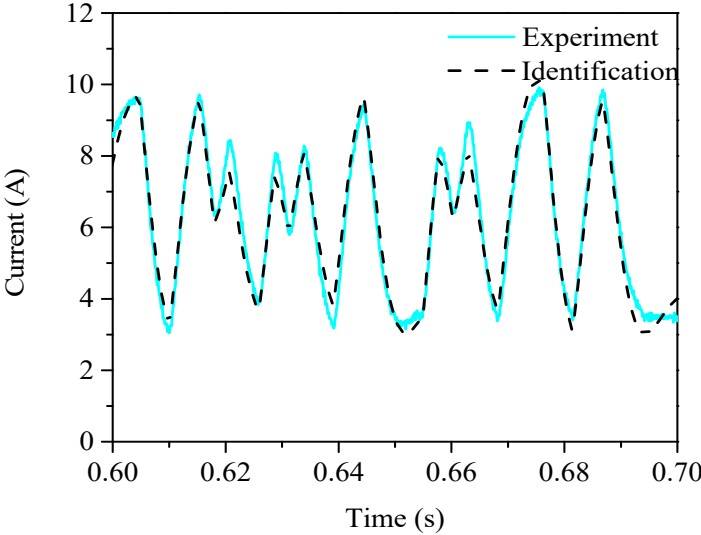

**Figure 7.** Comparison of the experimental and identification results at sampling rate of 800 Hz.

Table 4 shows the values of RT2 and YIC identified by the SRIV identification algorithm under different identification conditions. Figure 8 illustrates the Bode plots of the identified motor at different sampling rates. The Bode plots with different elapsed clock times clock times are shown in Figure 9. When the sampling rate is 800 Hz, RT2 is almost equal to 1, and the YIC value is also the sufficient negative.

**Table 4.** RT2 and YIC under different identification conditions.

| Identification Results | | Evaluation Index | | |
|---|---|---|---|---|
| Sampling Rate (Hz) | Amplitude (%) | Clock Time (ms) | RT2 | YIC |
| 700 | 20 60 | 200 | 0.873 | −9.627 |
| 800 | 20 60 | 200 | 0.903 | −10.562 |
| 900 | 20 60 | 200 | 0.875 | −10.128 |
| 800 | −60 60 | 200 | 0.939 | −11.426 |
| 800 | −60 60 | 400 | 0.954 | −11.882 |
| 800 | −60 60 | 500 | 0.957 | −11.892 |
| 800 | −60 60 | 600 | 0.981 | −13.077 |

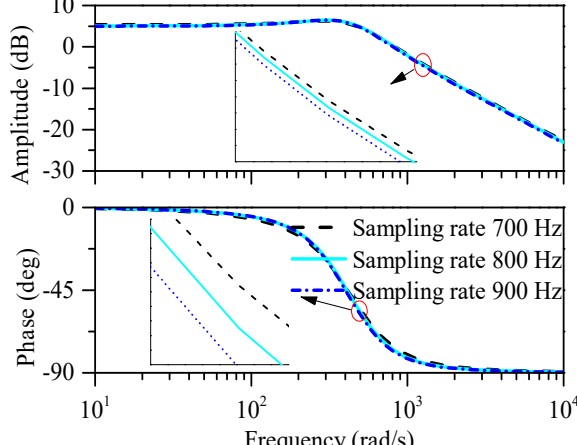

**Figure 8.** Bode plot of the identified motor under different sampling rates.

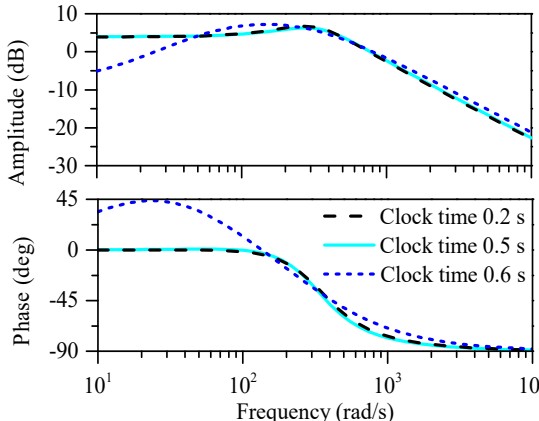

**Figure 9.** Bode plot of the identified motor under different clock times.

The evaluation parameters for the three different identification methods are listed in Table 5. Table 5 and Figure 10 show that the SRIV identification algorithm has the highest identification accuracy, which is consistent with the previous simulation results. The identification results obtained using different model structures are evaluated in Table 6. High RT2 and small YIC values are obtained when *nb* and *nf* are equal to 2.

Figure 11 shows a comparison between the test data and identification data, showing that the identification data are slightly different from the test data in the peak position. However, the trend of the figure is consistent with the test data, and the identification data coincided with the test data in other location.

**Table 5.** Evaluation parameters under different identification algorithms.

| Identification Algorithm | RT2 | MSE | FIT (%) |
|---|---|---|---|
| SRIV | 0.957 | 4.643 | 79.01 |
| IVSVF | 0.913 | 9.217 | 70.42 |
| LSSVF | 0.837 | 17.4 | 59.36 |

**Table 6.** Evaluation of the identification results using different model structures.

| *nb* | *nf* | *nk* | RT2 | YIC | Cond | AIC |
|---|---|---|---|---|---|---|
| 3 | 3 | 0 | 0.97269 | −12.621 | $1.6182 \times 10^6$ | 1.0582 |
| 1 | 1 | 0 | 0.91941 | −12.304 | 26.673 | 2.1396 |
| 2 | 2 | 0 | 0.98023 | −12.598 | 7255.8 | 1.5362 |
| 3 | 2 | 0 | 0.9628 | −11.754 | $8.3843 \times 10^5$ | 1.3774 |
| 1 | 2 | 0 | 0.93298 | −10.875 | 28,244 | 1.9562 |

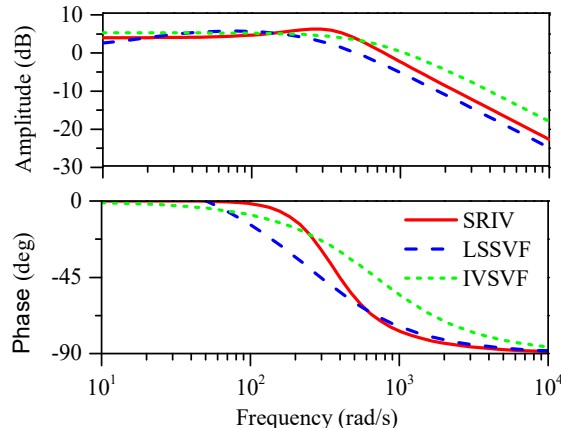

**Figure 10.** Bode plot using different identification algorithms.

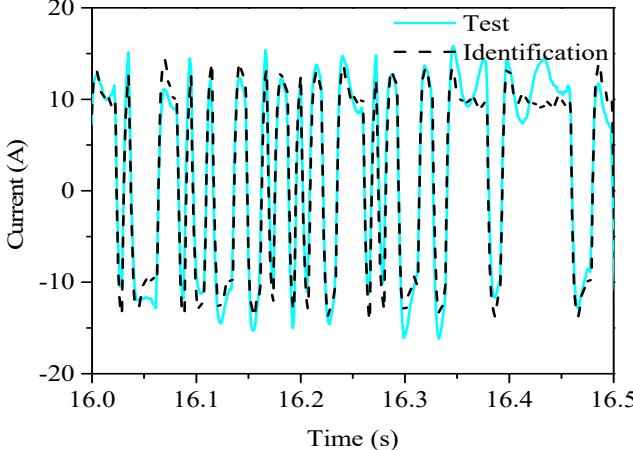

**Figure 11.** Comparison of the test data and identification data.

The current and voltage of the assist motor and the current of the load motor during the identification test are presented in Figure 12. The voltage and current of the assist motor exhibit the same trends, whereas the load current is always negative, which prevents the motion of the assist motor. This result indicates that the trend of the test data is correct.

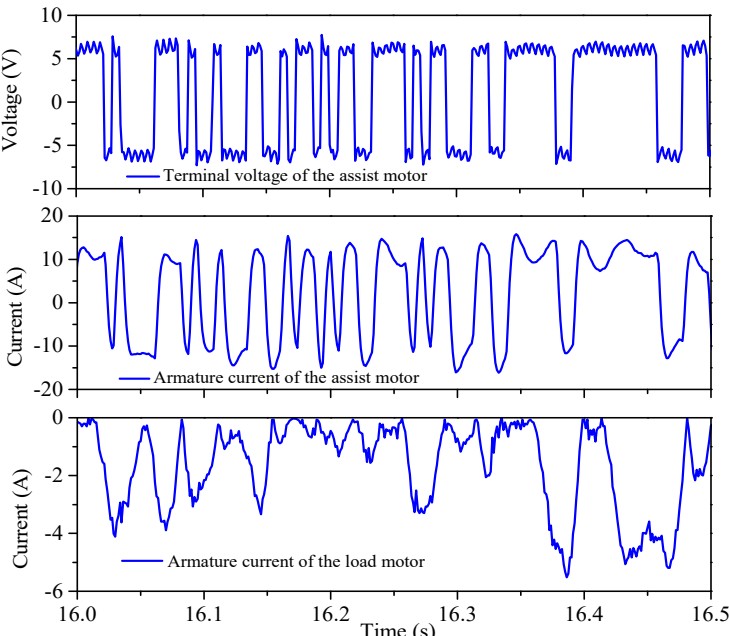

**Figure 12.** Time profiles of the assist motor current and voltage.

Figure 13 shows the verification of the identification model. The trend of the identification curve is largely consistent with Figure 13, indicating that the identification requirement is reached under the identification conditions of the SRIV, a sampling rate 800 Hz.

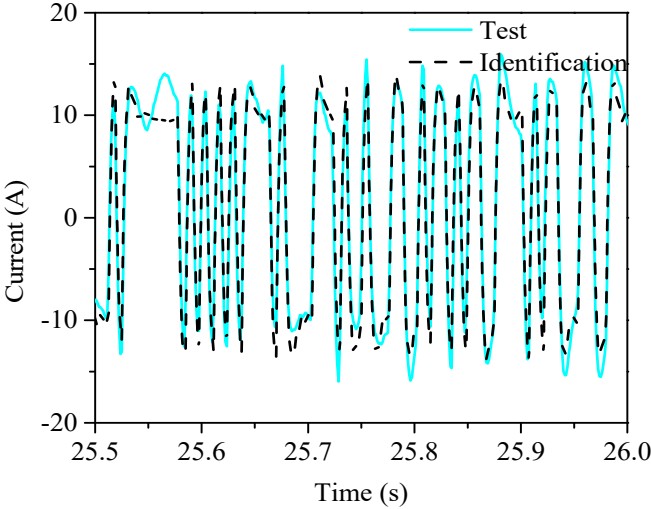

**Figure 13.** Verification of the identification model.

## 5. Identification Analysis and Test Bench Simulation of the EPS System

### 5.1. Identification Analysis of the EPS System

The identification simulation of the EPS is similar to the motor identification simulation. Because the input is the torque from the steering wheel and the output is the displacement of the rack,

the EPS system can be regarded as a single-input single-output (SISO) system. The input signal and sampling rate selection is consistent with the relevant contents in the previous section. Table 7 shows the parameter value used in the analysis of the EPS simulation. The transfer function of the EPS system is obtained from the previous section as shown in Equation (10). Combining Table 7, the transfer function of the EPS system is shown in Equation (30):

$$
\begin{aligned}
G_1(s) = \frac{p_r(s)}{T_d(s)} &= \frac{1}{\frac{r_p}{K_c}\left(J_c s^2 + B_c s + K_c\right)\left(M_r s^2 + B_r s + \frac{K_c}{r_p^2} + K_r\right) - \frac{K_c}{r_p}} \\
&= \frac{1.1518 \times 10^4}{s^4 + 29.438 s^3 + 6.497 \times 10^4 s^2 + 6.175 \times 10^5 s + 8.181 \times 10^6}
\end{aligned}
\tag{30}
$$

**Table 7.** EPS parameters.

| EPS Parameters | Value | Unit |
|---|---|---|
| Steering wheel moment of inertia $J_c$ | 0.04 | kg·m$^2$ |
| Torsional stiffness $K_c$ | 115 | N·m/rad |
| Steering wheel damping $B_c$ | 0.325 | N·m/(rad/s) |
| Rack and wheel assembly mass $M_r$ | 32 | kg |
| Rack damping $Br$ | 653.2 | N/(m/s) |
| Tire or rack centring spring rate $K_r$ | 91,061 | N/m |
| Pinion radius $r_p$ | 0.0071 | m |

Figure 14 shows the Bode plot of the EPS obtained using the SRIV identification algorithm at an SNR of 30 dB and different sampling rates. The identified model is the closest to the real model when the sampling rate is 800 Hz. The Bode plot illustrates that for a sampling rate of 70 Hz, the identified model is not suitable as the actual model.

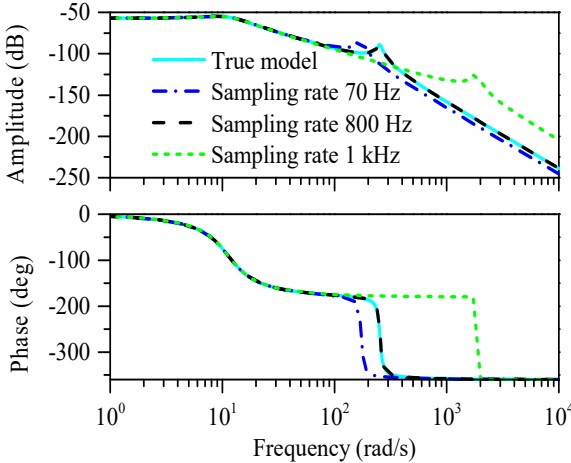

**Figure 14.** Bode plot of the EPS using different sampling rates.

Figure 15 shows the EPS Bode plot using different identification algorithms based on an SNR of 30 dB and a sampling rate of 800 Hz. The SRIV algorithm identification model clearly has the best identification resolution. The phase frequency characteristics of the IVSVF and LSSVF identification models deviate considerably from the phase frequency characteristics of the actual model.

Table 8 presents the evaluation of different model structures using the SRIV algorithm with an SNR of 30 dB and a sampling rate of 800 Hz. The evaluation parameters of the optimal model structure are also listed in Table 8. According to the data in Table 8, when $nb = 1$, $nf = 4$ and $nk = 0$, the structure of the model is the structure of the actual model. RT2 is the largest value, and the YIC value is the most negative ($-18.755$), indicating the identification conditions and the feasibility of the algorithm. Figure 16 shows the input signal of the identification simulation, which is the PRBS generated by

the 8-level shift register. The overlap of the coarse lines is caused by the high frequency of the PRBS. Figure 17 shows a comparison of the simulation and test data identified by the SRIV algorithm when *nb* = 1, *nf* = 4 and *nk* = 0.

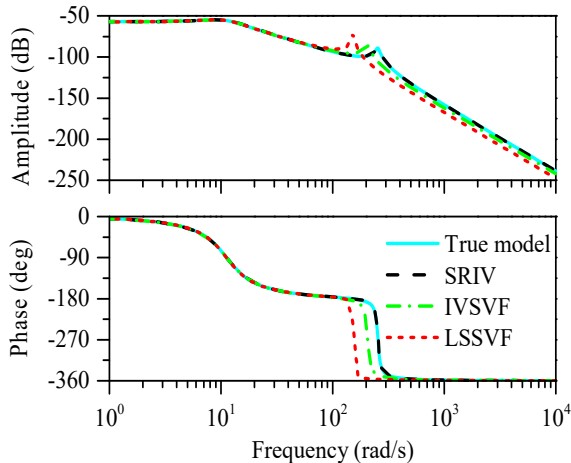

**Figure 15.** Bode plot of the EPS using different identification algorithms.

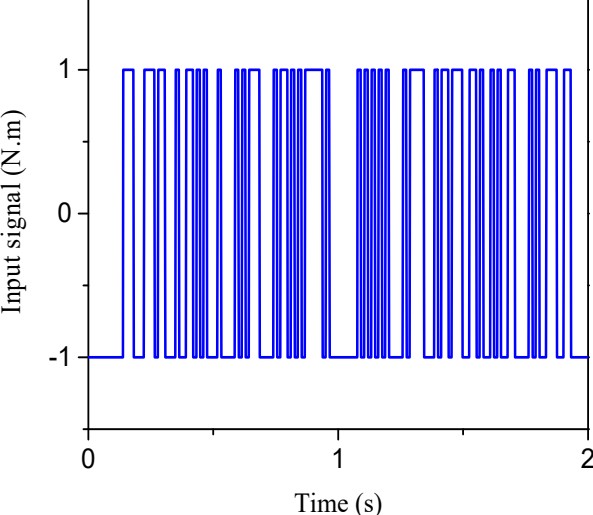

**Figure 16.** Input signal of the identification simulation.

**Table 8.** Evaluation of the different model structures.

| *nb* | *nf* | *nk* | RT2 | YIC | Cond | AIC |
|------|------|------|---------|---------|-------------------------|---------|
| 1 | 2 | 0 | 0.99843 | −16.755 | 3527.9 | −22.109 |
| 1 | 4 | 0 | 0.99886 | −18.755 | 78,870 | −22.425 |
| 4 | 3 | 0 | 0.99855 | −10.839 | $9.6734 \times 10^{10}$ | −22.184 |
| 2 | 2 | 0 | 0.99845 | −10.794 | 47,544 | −22.125 |
| 3 | 3 | 0 | 0.99859 | −10.478 | $3.8167 \times 10^{8}$ | −22.21 |

As shown in Figure 17, there are more burrs in the curve of the test data. This trend is related to the sampling noise in the test. When the test is identified, the input signal must be filtered to eliminate the effect of noise on the test data. The simulation result of the SRIV algorithm is relatively smooth, and the degree of coincidence between them is relatively high, illustrating a high identification resolution.

Through the simulation analysis of the EPS model, the SRIV identification algorithm obtains a good identification result under the simulation conditions of an 800 Hz sampling rate and an

SNR of 30 dB. In the true model structure of the EPS system (*nb*, *nf*, *nk* are [1, 4, 0]), the key evaluation parameters of the model structure are superior. The simulation results demonstrate that the identification of the EPS is consistent with the actual situation and will also play a guiding role in the following experiments.

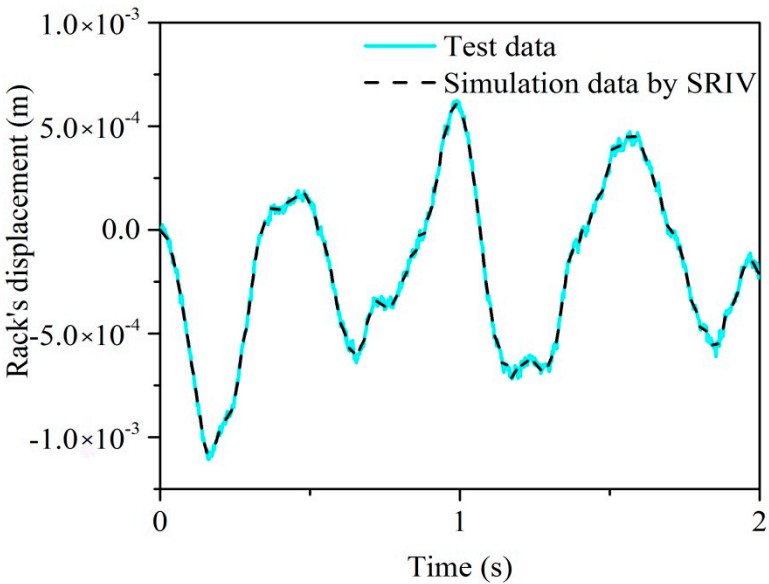

**Figure 17.** Comparison of the SRIV simulation data and test data.

*5.2. Identification Experiment for the EPS System*

For the identification experiment of the EPS, the input signal is the steering wheel torque provided by the driver. Unlike the motor identification experiment, the input signal of the motor is provided by programming the PRBS. Hence, when the driver is manipulating the steering wheel, the torque frequency and amplitude should be varied over a large range to ensure the continuity of the input signal. The input torque from the driver varies linearly with the output voltage signal of the torque sensor. In the experimental process, the data acquisition system is used to collect and save the current of the voltage signal of the torque main, assist motor and current of the load motor. The voltage range of the main and sub-main output of the torque sensor is 0.5 V. When the steering wheel has no input torque, the main and sub-main voltage is 2.5 V; when the steering wheel input torque is positive, the main voltage is greater than 2.5 V; and when the input torque of the steering wheel is negative, the main voltage is less than 2.5 V.

In contrast to the identification simulation of the EPS system, the input torque and rack displacement in the EPS identification experiment cannot be measured directly. Hence, the system structure of the actual model will also differ from the theoretical analysis of the structure of simulation model [1 4 0]. Based on transfer function of the EPS system in Equation (10) combining the experiment data, the EPS system parameters include torsional stiffness, steering wheel damping and rack damping, which affect significantly response of the system. It is a very necessary to identify these parameters. The structure of the EPS by experimental identification can be obtained by comparing the RT2 and YIC values of different system structures at different sampling rates and selecting the best system structure. The SRIV identification algorithm is used at sampling rates of 100 Hz, 500 Hz, 800 Hz, 900 Hz and 1 kHz. The structure of the actual model [3 4 0] appears in the optimal system structure. Table 9 shows the evaluation using the [3 4 0] structure and different sampling rates. According to Table 9, the value of YIC is the most negative when the sampling rate is 800 Hz. In contrast, the value of RT2 is the largest when the sampling rate is 800 Hz. According to the comparative analysis of data, the structure [3 4 0] is

selected as the main structure of the EPS system with a sampling rate of 800 Hz. The results show the torsional stiffness ($K_c = 103.6$), steering wheel damping ($B_c = 0.438$) and rack damping ($B_r = 702.4$).

Figure 18 shows the step response of the EPS system obtained at sampling rates of 800 Hz and 1 kHz, where the steady-state response of the system is a constant negative value. When the input torque of the system is stable, the speed of the load motor and the current produced is constant, which corresponds with the response of the actual system. Table 10 shows the evaluation parameters for three different identification algorithms, illustrating that the SRIV identification algorithm has the highest identification resolution, followed by the IVSVF identification algorithm and then the LSSVF algorithm.

**Table 9.** Evaluation parameters at different sampling rates.

| Structure of the EPS System | Sampling Rate (Hz) | RT2 | YIC |
|---|---|---|---|
| | 100 | 0.975 | −9.990 |
| | 500 | 0.965 | −10.014 |
| $nb = 3$, $nf = 4$, $nk = 0$ | 800 | 0.984 | −12.265 |
| | 900 | 0.981 | −12.251 |
| | 1000 | 0.981 | −11.155 |

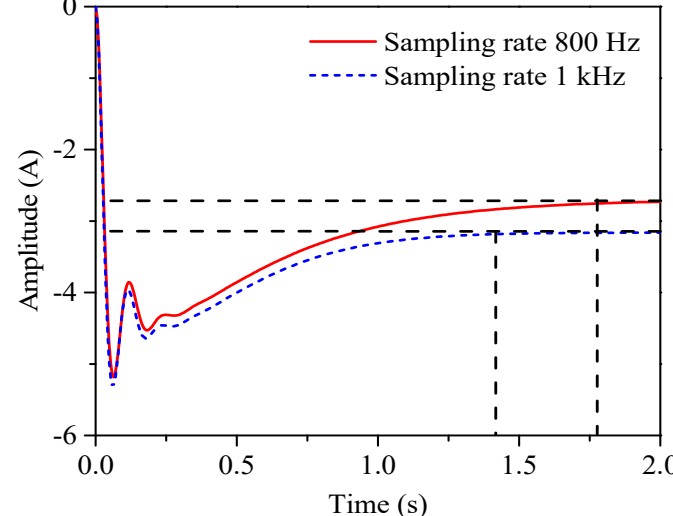

**Figure 18.** Step response of EPS for sampling rates of 800 Hz and 1 kHz.

**Table 10.** Evaluation parameters using three different identification parameters.

| Identification Algorithm | RT2 | MSE | FIT (%) |
|---|---|---|---|
| SRIV | 0.981 | 0.204 | 86.28 |
| IVSVF | 0.970 | 0.475 | 79.07 |
| LSSVF | 0.929 | 0.765 | 73.43 |

Figure 19 illustrates the EPS Bode plot obtained using different identification algorithms under an 800 Hz sampling rate. The Bode plot of the LSSVF identification algorithm fluctuates at a low frequency. The evaluation parameters in Table 10 illustrate that the highest identification accuracy is the SRIV identification algorithm.

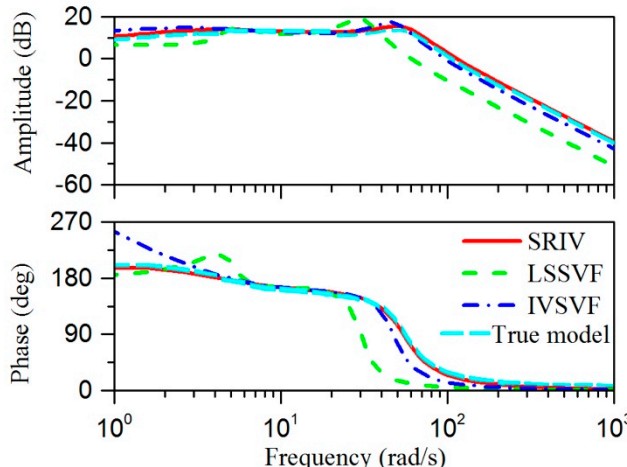

**Figure 19.** Bode plot of the EPS using different identification algorithms.

Through the comparisons and analysis of the different identification conditions and identification results, the optimal identification algorithm and identification conditions are the SRIV identification algorithm and a sampling rate of 800 Hz. Figure 20 shows the identification of the input and output signals for the system, where the change in the input torque is reflected in the change in the main voltage of the torque sensor. The line in the starting position shows no input torque at this time, and the main and sub-main voltage is 2.5 V. The amplitude and frequency of the main road vary considerably, satisfying the requirement of the identifying input signal. Unlike the input signal of the motor identification experiment, the EPS identification input signal is from the driver turning the steering wheel. Therefore, the identification input signal is different each time. Data from several EPS identification experiments are needed to adequately identify the system.

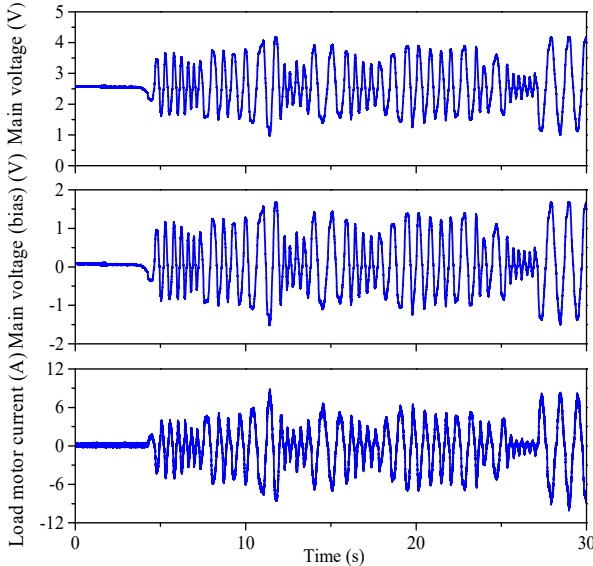

**Figure 20.** Input and output of the EPS identification system.

The identification data and experiment data coincide except for the peak position. Hence, the EPS model was identified by the SRIV algorithm at a sampling rate of 800 Hz. Figure 21 presents the verification result of the EPS identification model. The input signal between 23 and 28 s is considered in the identification model to obtain the identification data. The identification data is compared with the output signal of the original test to verify the accuracy of the identification model.

The identification data even follow the trend of the test data in the peak and trough positions, which further confirms that the identified EPS model produces highly accurate identification results using the SRIV identification algorithm and a sampling rate of 800 Hz.

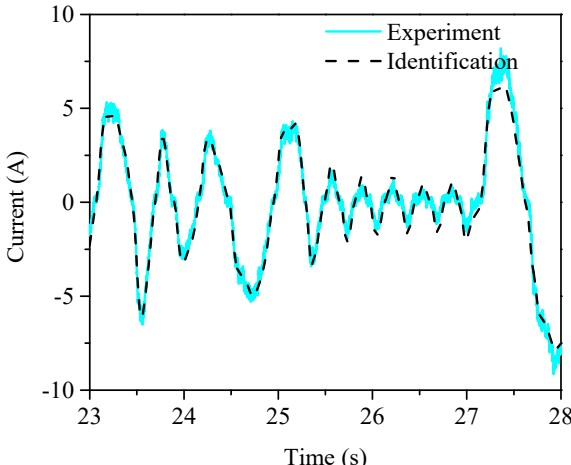

**Figure 21.** Verification of the identified EPS system.

The aim of this paper is to design a control algorithm for the EPS system that ensures that the curve of the desired assist characteristic tracks the curve of the actual assist characteristic at different vehicle velocities. This curve is established according to the correlation between the drive torque ($T_d$) and vehicle speed (*v*), which is obtained from the vehicle manufacturer. Figure 22 shows the assist torque characteristic curves versus the driver torque with respect to different vehicle speeds. Figure 22 illustrates that the assist torque is proportional to the steering wheel torque, with the assist torque ($T_a$) increasing with decreasing vehicle velocity to ensure maneuverability. In contrast, when the vehicle velocity increases, the assist torque decreases while maintaining the system stability.

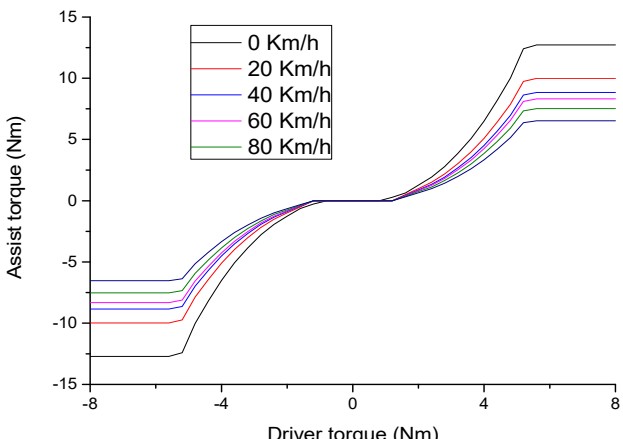

**Figure 22.** Correlation of the assist torque and driver torque at different vehicle speeds.

## 6. Controller Design

In this section, the design process of the loop-shaping controller is described. The diagram of the loop-shaping controller is shown in Figure 23, where *r*, *e*, $\hat{K}$, *u*, $d_i$, $u_t$, *G*, *d*, *y*, *n* are the desired reference signal, the control error between the desired reference and feedback signals, the controller, the controller output, the disturbance input, the control plant, the output disturbances, the output of the system and the measured noise, respectively.

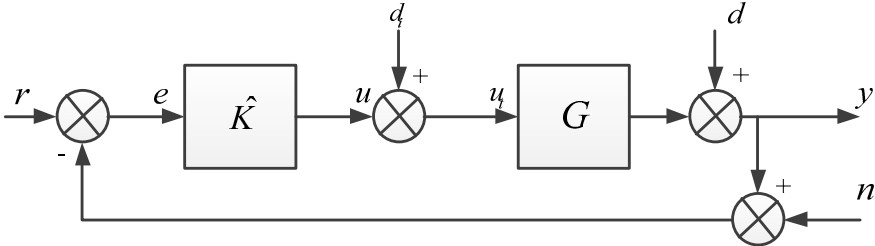

**Figure 23.** Loop-shaping controller diagram.

The design technique of the loop-shaping control is based on certain functions used to evaluate the stability of the system, including the sensitivity function, complementary sensitivity function and loop-shaping gain. These functions are defined as follows:

The sensitivity function: $S = \dfrac{1}{1 + G\hat{K}}$

The complementary sensitivity function: $T = \dfrac{G\hat{K}}{1 + G\hat{K}}$. The loop-shaping gain: $L = G\hat{K}$

The transfer function of the closed-loop system, which has the following relationships, is shown in Figure 23:

$$y = Tr - Tn + GSd_i + Sd \tag{31}$$

$$u = \hat{K}Sr - \hat{K}Sn - Td_i - \hat{K}Sd \tag{32}$$

$$u_t = \hat{K}Sr - \hat{K}Sn + Sd_i - \hat{K}Sd \tag{33}$$

$$e = Sr - Sn - GSd_i - Sd \tag{34}$$

The quality objectives of the desired system requirement can be proposed based on Equations (32)–(34). For example, the sensitivity function ($S$) must be small to reduce the influence of the output disturbances. Similarly, the complementary sensitivity function ($T$) must also be small to reduce the influence of the measured noise on the output signal ($y$). The constraint between the sensitivity function ($S$) and complementary sensitivity function ($T$) is satisfied by $S + T = 1$, indicating that the sensitivity function and complementary sensitivity function cannot be reduced simultaneously. Hence, the system is less affected by disturbances ($d$) and ($d_i$). Therefore, $|S|$ and $|GS|$ or ($|S|$ and $|\hat{K}S|$) must be small in the lower frequencies. Then, one obtains $|G\hat{K}| - 1 \leq |1 + G\hat{K}| \leq |G\hat{K}| + 1$. Thus, if $|G\hat{K}| > 1$, $\dfrac{1}{|G\hat{K}|+1} \leq |\dfrac{1}{1+G\hat{K}}| \leq \dfrac{1}{|G\hat{K}|-1}$, or if $|L| > 1$, $\dfrac{1}{|L|+1} \leq |S| \leq \dfrac{1}{|L|-1}$. From these facts, one obtains $|S| \ll 1 \Leftrightarrow |L| \gg 1$.

In addition, if $|L| \gg 1$, $|GS| = \left|\dfrac{G}{1+G\hat{K}}\right| \approx \dfrac{1}{|\hat{K}|}$, $|\hat{K}S| = |\dfrac{\hat{K}}{1+G\hat{K}}| \approx \dfrac{1}{|G|}$.

From the above analysis, it is reasonable to propose that the EPS system must satisfy the conditions $|G\hat{K}| \gg 1$, $|\hat{K}| \gg 1$ to ensure the quality objective in the low frequency domain $(0, \omega_l)$. The variable $(\cdot)_l$ refers to the low frequency.

To ensure stability and adequate disturbance rejection ability in the high-frequency domain $(\omega_l, \infty)$, the system must satisfy the conditions $|G\hat{K}| \ll 1$, $|\hat{K}| \ll M$. The loop-shaping gain technique is a trade-off technique between control performance and system stability. Hence, the system requirement must achieve high control performance in the lower frequency domain and system stability in the high-frequency domain, as illustrated in Figure 24. The diagram of the loop-shaping controller for the EPS system is shown in Figure 25.

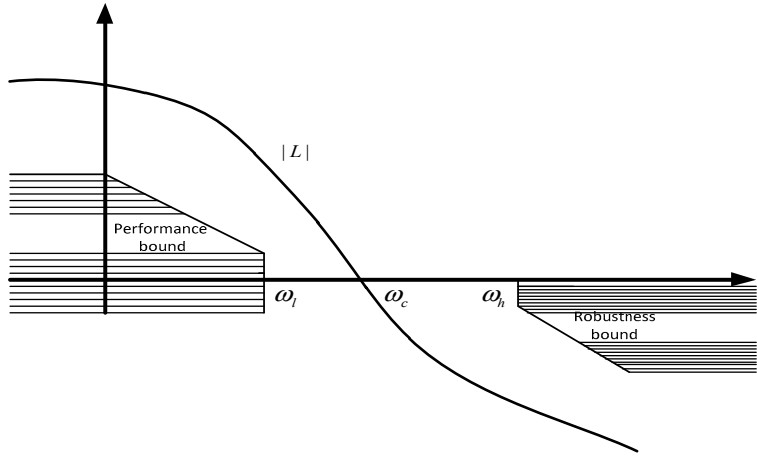

**Figure 24.** Desired loop gain.

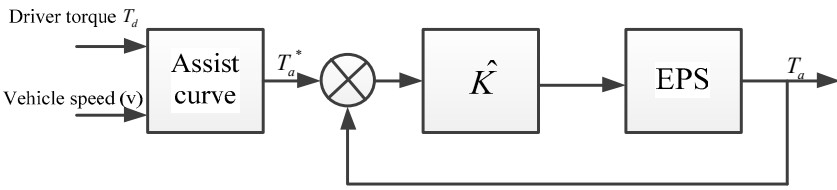

**Figure 25.** Diagram of the loop-shaping controller for EPS.

## 7. EPS Test Bench

In the test bench of the EPS system, the required test equipment includes the EPS prototype, current sensor and data acquisition card. The overall block diagram of the EPS identification experiment platform is shown in Figure 26.

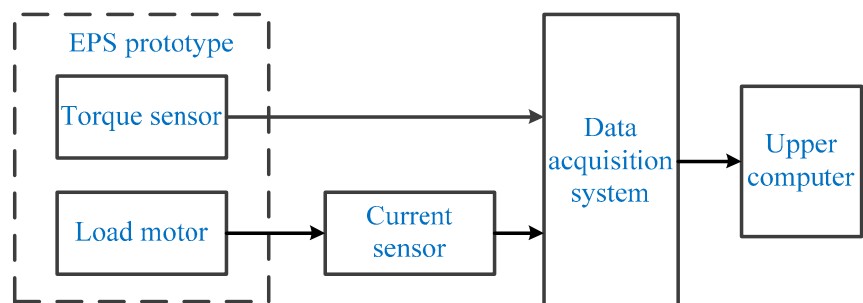

**Figure 26.** Overall block diagram of the EPS electric motor parameter identification experiment platform.

The main equipment for the verification experiment of the EPS control algorithm is the EPS prototype, dSPACE real-time simulation platform, TTL level conversion board, EPS drivers and a power supply. The overall block diagram of the verification experiment platform is shown in Figure 27. The real-time simulation platform of the dSPACE card includes the hardware and software platforms. The hardware part contains the real-time processor and I/O interface, in which the real-time processor can compile directly from the MATLAB/Simulink and the I/O interface contains the A/D, D/A, PWM signals. The software mainly contains the Real-Time Interface (RTI), Control Desk, and Target Link. The voltage range of the power supply for the I/O port of the dSPACE is 8–18 V. The EPS prototype consists mainly of the assist motor, load motor, torque sensor, and steering wheel. The positive and negative poles of the load motor are short-connected. When the steering wheel rotates, it provides resistance for the EPS system. Figure 28 shows the EPS bench.

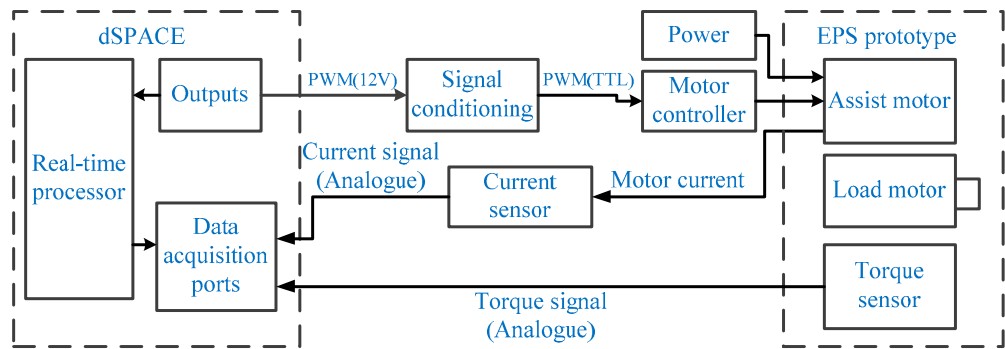

**Figure 27.** Block diagram of the verification experiment platform for the EPS control algorithm.

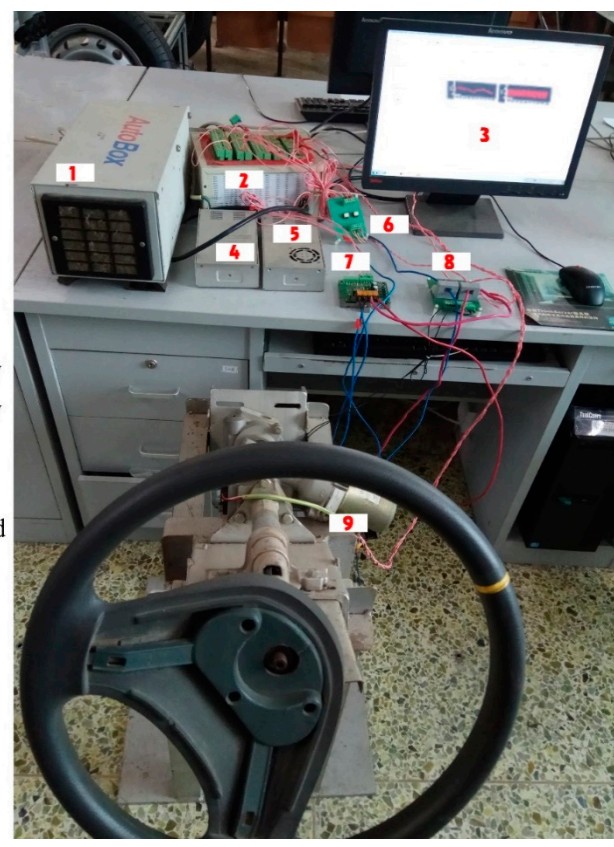

1. dSPACE
2. Connector
3. Upper computer
4. 12 V power supply
5. 24 V power supply
6. Motor driver
7. Current sensor
8. Conditioning board
9. EPS prototype

**Figure 28.** EPS test bench.

## 8. Verification of the EPS Control Algorithm

In this section, two control algorithms, namely, the PID control algorithm and the proposed loop-shaping control algorithm, are applied to control the assist motor in the assistance mode of the EPS. To implement this in real time for the EPS system, the continuous time model in the Laplace s domain must be transferred to a discrete time model in the z domain.

The setup procedure of the control algorithm includes three steps:

**Step 1.** Establish the new model based on MATLAB/Simulink and configure the RTI interfaces, including the common I/O, PWM and AD modules.

**Step 2.** Establish the control algorithm model for the EPS system, filter the collected analogue signals and compile and generate dSPACE executable SDF files.

**Step 3.** Build the new experimental project in the Control Desk software of dSPACE. The generated SDF files are downloaded into the real-time dSPACE card via the Ethernet network.

Finally, the overall EPS frame system is controlled in real time via the control panel of Control Desk.

*Case 1.* *PID control algorithm*

For the PID control algorithm, the trial-and-error method is utilized to design the I and P parameters, and the D parameter is set to 0. When the parameters P and I are adjusted, a step target current is given to the control system, and thereafter, the P value is adjusted to make the actual current value slightly higher than the target current value but without too large an overshoot. The actual current reaches the target current and then undergoes a certain attenuation, which indicates that the control requirements cannot be met solely by adjusting the P value; the I value must also be adjusted. While adjusting the I value, the P value must also be adjusted accordingly. This conflict is avoided by iteratively adjusting P and I until the demanded control effect is achieved. When the target torque of the motor is less than 15 N·m, P is 0.054 and I is 0.0006.

*Case 2.* *Proposed loop-shaping control algorithm*

As previously discussed, The SRIV algorithm has the highest estimation accuracy, which is used to obtain a nominal model for the EPS system. The structure [3 4 0] is the model of EPS system $G = \frac{-1.138 \times 10^{11}s^2 + 2s + 0.4}{3s^4 + 278.51s^3 + 1.1692 \times 10^6 s^2 + 2.937 \times 10^7 s + 2 \times 10^{10}}$ based on the identification experiment and the controller is $\hat{K}$, as shown in Figure 23. However, this model is not absolutely accurate. The EPS system needs the controller to enhance the stability of the EPS system. Hence, from the above analysis of the controller design, the loop-shaping controller design procedure is summarized in four main steps:

**Step 1:** Select a pre-compensator $W_1$ and post-compensator $W_2$. These two shaping functions are added to generate the shaped plant $G_s$, which is written as follows:

$$G_s = W_2 G W_1 \tag{35}$$

In SISO systems, the weighting function $W_1$ and $W_2$ can be chosen as

$$W_1 = 9,05 \frac{s + 956}{s + 0.32} \text{ and } W_2 = 1 \tag{36}$$

where $W_2$ can be chosen as a constant, since the effect of sensor noise is negligible. In this method, the shaped plant is formulated as normalized co-prime factor, which separates the shaped plant $G_s$ into normalized nominator $N_s$ and denominator $M_s$ factors. If the shaped plant $G_s = N_s M_s^{-1}$, the perturbed plant is written as

$$G_\Delta = (N_s + \Delta_{Ns})(M_s + \Delta_{Ms})^{-1} \tag{37}$$

where $\Delta_{Ns}$ and $\Delta_{Ms}$ are stable and unknown, representing the uncertainty satisfying $\|\Delta_{Ns}, \Delta_{Ms}\| \leq \varepsilon$, $\varepsilon$ is the uncertainty boundary called the stability margin.

**Step 2:** Given a shaped plant $G_s$ and $A$, $B$, $C$, $D$ represent the shaped plant in the state-space form. To determine $\gamma_{\min}$, there is a unique method, as follows [33].

$$\gamma_{\min} = (1 + \lambda_{\max}(Z_s X_s))^{1/2} \tag{38}$$

where $\lambda_{\max}$ is the maximum eigenvalue and $X_s$ and $Z_s$ are the solutions of two Riccati, as:

$$(A - BS^{-1}D^T C)Z_s + Z_s(A - BS^{-1}D^T C)^T - Z_s C^T R^{-1}CZ + BS^{-1}B^T = 0 \tag{39}$$

$$(A - BS^{-1}D^T C)^T X_s + X_s(A - BS^{-1}D^T C) - X_s BS^{-1}B^T X + C^T R^{-1}C = 0 \tag{40}$$

where

$$S = I + D^T D$$

$$R = I + DD^T$$

To ensure the robust stability of the nominal plant, the weighting function is selected so that $1 < \gamma_{\min} < 5$ [33]. If $\gamma_{\min}$ is too large, then return to step 1 and change $W_1$, $W_2$.

**Step 3:** Choose $\gamma > \gamma_{\min}$; the $\hat{K}_\infty$ controller must satisfy the following equation [13]:

$$\left\| \begin{bmatrix} \hat{K}_\infty \\ I \end{bmatrix} (I - G_s \hat{K}_\infty)^{-1} M_s^{-1} \right\| \leq \gamma \tag{41}$$

**Step 4:** The controller is synthesized as follows:

$$\hat{K} = W_1 \hat{K}_\infty W_2 = \frac{372.3s^4 + 6714s^3 + 67960s^2 + 55790s + 11950}{s^5 + 40s^4 + 709.3s^3 + 6310s^2 + 2648s + 2.641} \tag{42}$$

Moreover, we established the control algorithms of the EPS based on MATLAB/Simulink and the control panel of Control Desk.

The results of the reference assist torque step response with an amplitude of 1 Nm are shown in Figure 29. The obtained values for rise time $t_{ris}$, maximum overshoot $o_{max}$, and settling time $t_{set}$ enable an objective comparison of the different controllers in Table 11.

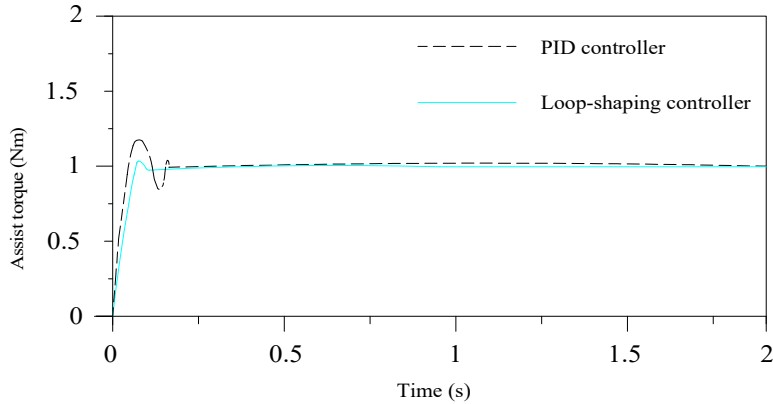

**Figure 29.** Comparison of the assist torque response using the PID controller and loop-shaping controller.

**Table 11.** Results of the 1 Nm reference assist torque response.

| Controller | $O_{max}$ (%) | $t_{ris}$ (s) | $t_{set}$ (s) |
|---|---|---|---|
| **PID controller** | 23.4 | 0.056 | 0.178 |
| **Loop-shaping controller** | 8.7 | 0.064 | 0.075 |

Figure 29 shows a comparison of the assist torque response under the PID controller and loop-shaping controller. The results obtained with the proposed loop-shaping controller are very good, because the settling time and the overshoot time are very small. Although the PI controller shows a fast response, its behavior is not satisfied, because it oscillates strongly. Consequently, the proposed loop-shaping controller reduces the disturbances for the EPS system, and it provides a good driver feel during the steering process.

Figure 30 is an actual assist torque at 40 km/h speed under the PID controller. From the figure it can be seen that the actual torque tracks the desired torque, but the assist torque characteristic does not have enough stability. Its behavior is not satisfied, because it oscillates strongly. Figure 31 is an actual assist torque at 20 km/h speed under the loop-shaping controller. It shows that the actual assist torque is good tracking the desired assist torque. Furthermore, the assist torque characteristic is high stability without overshoot.

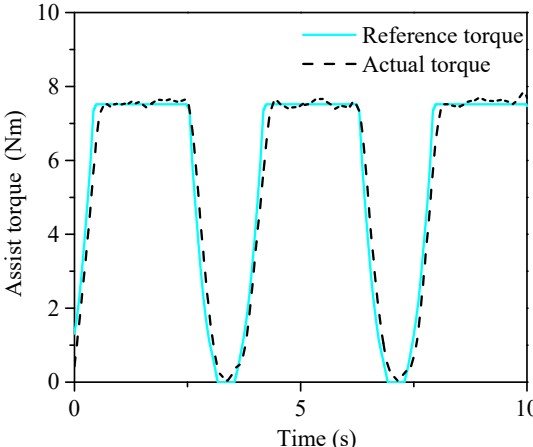

**Figure 30.** Actual torque at 40 km/h velocity under the PID control algorithm.

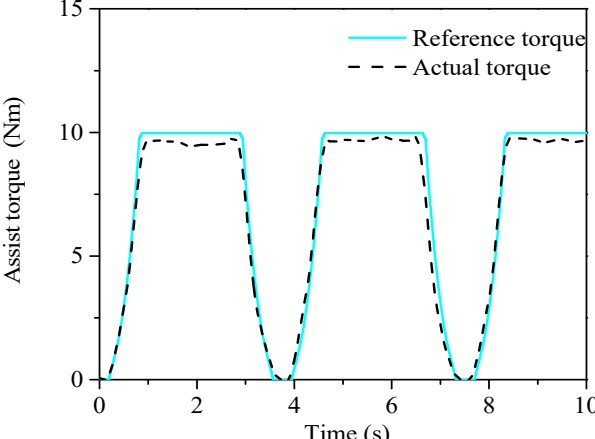

**Figure 31.** Actual torque at 20 km/h velocity under the proposed control algorithm.

Figure 32 shows actual assist characteristic curve under the PID control algorithm with different velocities. Although the actual assist torque characteristic tracks the desired assist torque, the desired assist torque characteristic oscillates strongly. Figure 33 shows actual assist characteristic curve using the loop-shaping controller with different velocities. The actual assist torque characteristic not only tracks the desired assist torque characteristic curve but also ensures system stability because it oscillates smoothly.

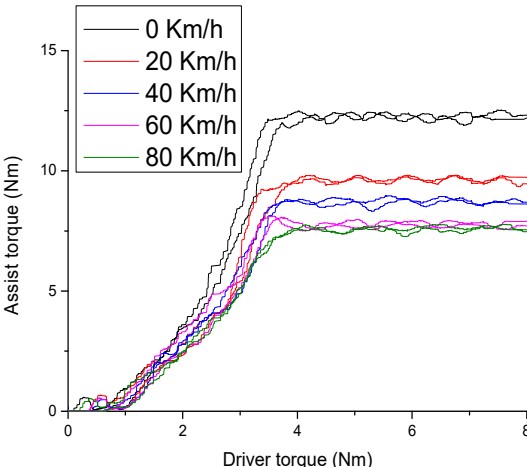

**Figure 32.** Actual assist characteristic curve under the PID control algorithm with different velocities.

The results obtained above demonstrate that although the PID control algorithm tracks the desired assist torque characteristic curve, the assist torque characteristic curve does not have enough stability. The proposed loop-shaping control algorithm not only tracks the desired assist torque characteristic curve but also ensures system stability. Thus, the assist characteristic curve obtained using the proposed loop-shaping control algorithm the disturbances are rejected to provide good tracking performance and ensure good steering feel.

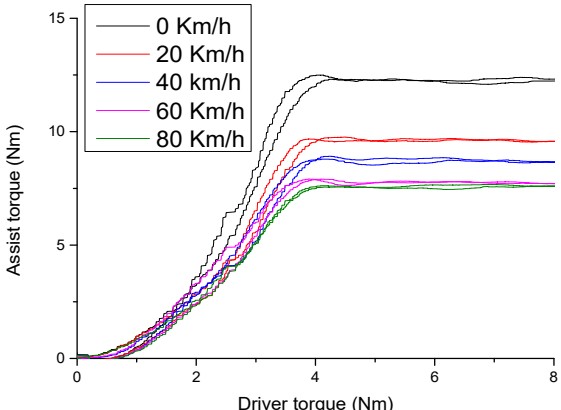

**Figure 33.** Actual assist characteristic curve using the proposed loop-shaping control algorithm with different velocities.

## 9. Conclusions

When the system is working, the system parameters usually change in a manner that reduces the model mismatching between the theoretical EPS models and actual EPS models. The identification of the EPS model parameters has also been discussed. Three identification algorithms, namely, SRIV, LSSVF and IVSVF, are applied to identify the parameters of the EPS system. Among the proposed algorithms, the SRIV algorithm is adopted for estimating the parameters of the control plant due to the highest estimation accuracy. Furthermore, in order to enhance control performance, the loop-shaping technique has been introduced to design the EPS controller system. The proposed loop-shaping controller provides the system stability and control performance of an EPS system. In addition, the effectiveness of the loop-shaping controller is confirmed, with both the classical PID and loop-shaping controllers applied to the test bench of the EPS system. The experimental results verify that the illustrated loop-shaping controller not only tracks the desired assist torque characteristic curve but also ensures the stability of the system.

**Author Contributions:** V.G.N. and X.G. conceived and designed the proposed approach; V.G.N. and C.Z. performed the experiments; V.G.N. and X.K.T. verified the proposed approach; V.G.N. wrote the manuscript; V.G.N., X.G., C.Z. and X.K.T. revised the manuscript.

**Funding:** This research received no external funding.

**Acknowledgments:** This work was supported in part by the National Natural Science Foundation of the People's Republic of China (Grant number: 51505354) and the Fundamental Research Funds for Central Universities. We would like to acknowledge the financial support for visiting scholars from the Chinese Scholarship Council in the form of grant numbers 2015GXZ070.

**Conflicts of Interest:** The authors declare no conflict of interest.

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
