# Peer review of "Parameter Estimation, Robust Controller Design and Performance Analysis for an Electric Power Steering System"

_algorithms, doi:10.3390/a12030057_

Round 1
Reviewer 1 Report
Electric power steering (EPS) identification and control system application-oriented paper is presented. Theorical and practical conditions are included as such, the assist motor and EPS actual parameters, sensor-measurement noise, and random road disturbance factors. Parameter estimation off-line are applied for get the actual system parameters. Simplified refined instrumental variable (SRIV), least squares state variable filter (LSSVF) and instrumental variable state variable filter (IVSVF) algorithms performance are compared, in order to select the best algorithm in EPS parameters estimation application with good performance results. The proposal control design approach is based in loop-shaping control with stability analysis through sensitivity function. Finally, PID and proposed loop-shaping controllers are compared.
The paper is presented in two parts parameter estimation and controller design, however is not evident the contributions related with parameter estimation and proposal control individually. There is inconsistence in parameter identification results, Nyquist theorem establish the minima sampling rate, in other words, as greater sampling rate better discrete model performance. The parameters estimation algorithms are defined in discrete time system, while theorical models are development in continuous time system, the relationship between discrete time model and continuous time model parameters are no analyzed. The proposal loop-shape control seems to be a particulate case of the PID controller.
1. In equations in lines 130 and 131 the parenthesis is open both no close.
2. Sequence is duplicated in line 263 it is included on PRBS, please review in all document.
3. Check the units of the pulse width modulation (PWM) duty cycle (200 ms) on line 270. According to what has been described, the response of the system reaches in the stable state (20 ms) before one cycle of the PWM ends (entry to the system).
4. Not all parameters an unknown and not all parameters change under operational conditions, please specified which model parameters are unknown and which parameters change with road disturbances.
5. In table 2 nb, nf, nk are not defined.
6. In the parameters estimation, in general, the sampling time must be independent of the system parameters when the Nyquist theorem is fulfilled, in table 3 (motor parameters estimation) justify why the parameters change when the sampling time changes.
7. Include aliasing effect analysis in Poison’s filters application.
8. As the precision of the motor parameters estimation of the EPS system is used in the controller design.
9. In figure 19 include real model trace in order to observe and compare models identification performance (see figure 2).
10. In figure 29 are shown PID and proposal loop-shaping control, the PID parameters (P, I and D gains) are calculated in lines from 533 to 541 but the loop-shaping parameter (K gain) is no reported. Please include the K value used in loop-shaping controller in figure 29.
11. There is performance analysis of proposal EPS control algorithm in section 8 figures 30, 31, 32 and 33, please include parameter estimation precision analysis in a controller performance in order to observe the parameter estimation algorithm benefits in controller.
12. The proposal loop-shape control is single input single output (SISO) system (defined in lines 333-334), then K controller in figure 25 is particulate case of PID controller considering I=0 and D=0, please include a brief analysis and comparison about the controllers structure.
Author Response
Dear Reviewer,
We thank the Reviewer very much for your time and your consideration. We have tried our best to revise the manuscript and we hope that it is satisfactory to the Reviewer. We are looking forward to receiving your clear comments and instructions this time.
Thank you very much!
We look forward to hearing from you soon.
Yours sincerely,
Xuexun Guo
School of Automotive Engineering, Wuhan University of Technology, Wuhan, China
On behalf of all authors

Reviewer 2 Report
The authors try to estimate several parameters of a electric power steering (EPS) system and design a controller for tracking performance. However, the paper is written carelessly and the contributions are not clearly clarified.
It is not clear how the identified model works in conjunction with the controller.
Some symbols are confusing or not defined, such as \Omega_c(s) in(8). The value of parameter k_e in (5) is not given.
The expression in (14) seems wrong, since it is impossible to derive y(t) according to the dimension of (14) and (15).
When deriving (9) from (3), where do the \theta_m and N go?
Using the definition of H^{\gamma}, the vectors in (14) are related to differentiations of y and u. How to measure the vector values?
What is M in line 491? In section 6, it is derived that the controller needs to satisfy |\hat{K}|>>1 and |\hat{K}|<<M. How to ensure these conditions in (38) and how to choose M are not clear.
In the simulation of section 4.1, what kind of noise has been used? How do you know that such noise reflects the practical parameters' variation due to load disturbances, measurement noise or other factors?
There are grammatical errors in line 70, 92, 155, 157, 160, 188, 484.
Author Response

(The authors gave the same response as above.)

Reviewer 3 Report
In this paper, a parameter identification and robust controller design have been presented to increase the tracking performance of the EPS system. Although methods work well, these are still some issues needed to be clarified. 1) The sampling rate is a key factor in discrete control system. The common law is the Nyquist theorem. It is common sense that the higher sampling rate, the more the memory space, the higher computational burden. Based on the reviewer’s understanding, it is unreasonable to compare the results of different sampling rate. Furthermore, what is the performance index to evaluate the system identification? 2) The performance of PI Controller is affected by the parameters of P & I. Although authors set values using the trial & error method, the P & I may not be optimal. The reason is that the steady state errors are fluctuated (shown in Figure 30) and not stable. Authors should give the reason for the axis’ scale difference between Figure 30 and Figure 31. The comparison is unfair under the different baseline. Furthermore, parameters’ value for the experiment should be presented and let readers understand the algorithm in detail. 3) There are some mistakes in formulations, for example, equation (14), (21), (27). Furthermore, some of symbols have been used several times for different physical meaning. 4) The structure of the manuscript is not good. Why did authors use three kinds of identification algorithms to estimate the parameters of ECS? Why did authors want to compare the loop-shaping controller to PID?
Author Response

(The authors gave the same response as above.)

Round 2
Reviewer 1 Report
Thanks for addressing my comments.
Author Response
Dear Reviewer #1,
We would like to express our thanks to the Reviewer #1 for your useful comments and your time to improve our paper. We have addressed all the comments and hope that this revised manuscript is acceptable for publication in Algorithms. All changes are highlighted in the manuscript and we have included our responses in this “Summary of Changes and Response to Reviewers’ Comments”.
Yours sincerely,
Xuexun Guo
School of Automotive Engineering, Wuhan University of Technology, Wuhan, China
On behalf of all authors

Reviewer 2 Report
The authors have addressed my concern.
Author Response
Dear Reviewer #2,
We would like to express our thanks to the Reviewer #2 for your useful comments and your time to improve our paper. We have addressed all the comments and hope that this revised manuscript is acceptable for publication in Algorithms. All changes are highlighted in the manuscript and we have included our responses in this “Summary of Changes and Response to Reviewers’ Comments”.
Yours sincerely,
Xuexun Guo
School of Automotive Engineering, Wuhan University of Technology, Wuhan, China
On behalf of all authors

Reviewer 3 Report
Although authors improved the manuscript according to the previous comments. There are still some issues need to be addressed.
1) As mentioned before, comparison between the PI and the proposed algorithm is unfair.
2) There were many mixed symbols in the manuscript. Furthermore, some symbols have not been given the definition.
3) Authors just claim the performance of proposed algorithm using figures which may not be enough. Authors should define some index to demonstrate the degree of the stability.
Author Response
Dear Reviewer #3,
We would like to express our thanks to the Reviewer #3 for your useful comments and your time to improve our paper. We have addressed all the comments and hope that this revised manuscript is acceptable for publication in Algorithms. All changes are highlighted in the manuscript and we have included our responses in this “Summary of Changes and Response to Reviewers’ Comments”.
Yours sincerely,
Xuexun Guo
School of Automotive Engineering, Wuhan University of Technology, Wuhan, China
On behalf of all authors

Round 3
Reviewer 3 Report
The required improvements have been answered.